# PIRLNav: Pretraining with Imitation and RL Finetuning for ObjectNav

**Ram Ramrakhya**[1]  **Dhruv Batra**[1,2]  **Erik Wijmans**[1]  **Abhishek Das**[2]

[1]Georgia Institute of Technology    [2]FAIR, Meta AI
[1]{ram.ramrakhya,dbatra,etw}@gatech.edu    [2]abhshkdz@meta.com

## Abstract

We study ObjectGoal Navigation – where a virtual robot situated in a new environment is asked to navigate to an object. Prior work Ramrakhya et al. (2022) has shown that imitation learning (IL) using behavior cloning (BC) on a dataset of human demonstrations achieves promising results. However, this has limitations – 1) BC policies generalize poorly to new states, since the training mimics actions not their consequences, and 2) collecting demonstrations is expensive. On the other hand, reinforcement learning (RL) is trivially scalable, but requires careful reward engineering to achieve desirable behavior. We present PIRLNav, a two-stage learning scheme for BC pretraining on human demonstrations followed by RL-finetuning. This leads to a policy that achieves a success rate of $65.0\%$ on ObjectNav ($+5.0\%$ absolute over previous state-of-the-art).

Using this BC→RL training recipe, we present a rigorous empirical analysis of design choices. First, we investigate whether human demonstrations can be replaced with 'free' (automatically generated) sources of demonstrations, *e.g.* shortest paths (SP) or task-agnostic frontier exploration (FE) trajectories. We find that BC→RL on human demonstrations outperforms BC→RL on SP and FE trajectories, even when controlled for the same BC-pretraining success on train, and even on a subset of val episodes where BC-pretraining success favors the SP or FE policies. Next, we study how RL-finetuning performance scales with the size of the BC pretraining dataset. We find that as we increase the size of the BC-pretraining dataset and get to high BC accuracies, the improvements from RL-finetuning are smaller, and that $90\%$ of the performance of our best BC→RL policy can be achieved with less than half the number of BC demonstrations. Finally, we analyze failure modes of our ObjectNav policies, and present guidelines for further improving them. Project page: ram81.github.io/projects/pirlnav.

## 1 Introduction

Since the seminal work of Winograd Winograd (1972), designing embodied agents that have a rich understanding of the environment they are situated in, can interact with humans (and other agents) via language, and the environment via actions has been a long-term goal in AI Smith & Gasser (2005); Hermann et al. (2017); Hill et al. (2017); Chaplot et al. (2018); Anderson et al. (2018b); Jain et al. (2019); Das (2020); Abramson et al. (2020); Weihs et al. (2021b); Lynch et al. (2022). We focus on ObjectGoal Navigation Anderson et al. (2018a); Batra et al. (2020), wherein an agent situated in a new environment is asked to navigate to any instance of an object category ('find a plant', 'find a bed', *etc.*); see figure 2. ObjectNav is simple to explain but difficult for today's techniques to accomplish. First, the agent needs to be able to ground the tokens in the language instruction to physical objects in the environment (*e.g.* what does a 'plant' look like?). Second, the agent needs to have rich semantic priors to guide its navigation to avoid wasteful exploration (*e.g.* the microwave is likely to be found in the kitchen, not the washroom). Finally, it has to keep track of where it has been in its internal memory to avoid redundant search.

Humans are adept at ObjectNav. Prior work Ramrakhya et al. (2022) collected a large-scale dataset of $80k$ human demonstrations for ObjectNav, where human subjects on Mechanical Turk teleoperated virtual robots and searched for objects in novel houses. This first provided a human baseline on ObjectNav of $88.9\%$ success rate on the Matterport3D (MP3D) dataset Chang et al. (2017)[1] compared to $35.4\%$ success rate of the best performing method Ramrakhya et al. (2022). This dataset was then used to train agents via imitation learning (specifically, behavior cloning).

---

[1]On val split, for 21 object categories, and a maximum of 500 steps.

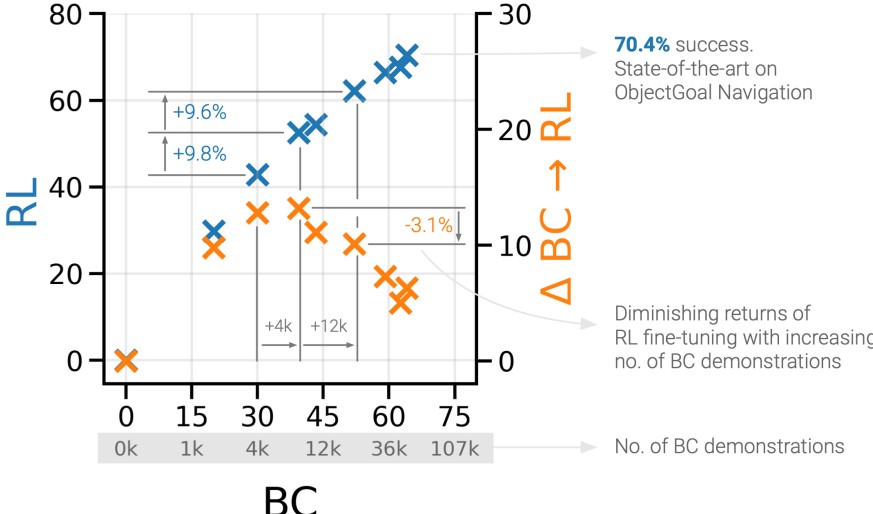

**Figure 1.** OBJECTNAV success rates of agents trained using behavior cloning (BC) *vs*. BC-pretraining followed by reinforcement learning (RL) (in blue). RL from scratch (*i.e*. BC=0) fails to get off-the-ground. With more BC demonstrations, BC success increases, and it transfers to even higher RL-finetuning success. But the difference between RL-finetuning *vs*. BC-pretraining success (in orange) plateaus and starts to decrease beyond a certain point, indicating diminishing returns with each additional BC demonstration.

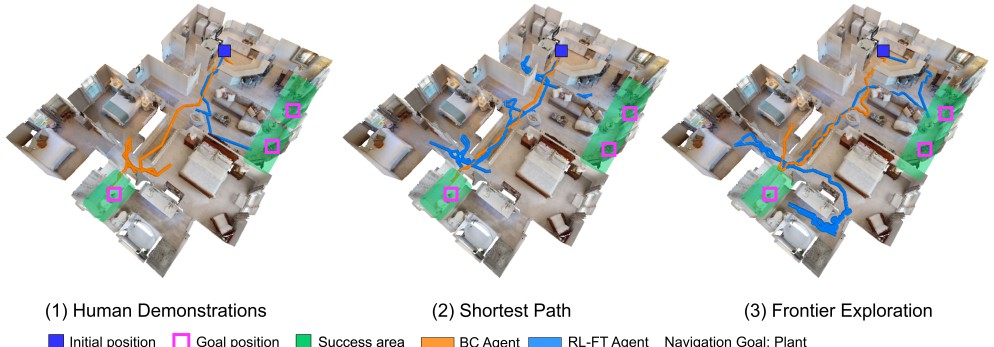

**Figure 2.** OBJECTNAV trajectories for policies trained with BC→RL on 1) Human Demonstrations, 2) Shortest Paths, and 3) Frontier Exploration Demonstrations.

While this approach achieved state-of-art results ($35.4\%$ success rate on MP3D VAL dataset), it has two clear limitations. First, behavior cloning (BC) is known to suffer from poor generalization to out-of-distribution states not seen during training, since the training emphasizes imitating actions not accomplishing their goals. Second and more importantly, it is expensive and thus not scalable. Specifically, Ramrakhya *et al*. Ramrakhya et al. (2022) collected $80k$ demonstrations on 56 scenes in Matterport3D Dataset, which took $\sim$2894 hours of human teleoperation and $50k$ dollars. A few months after Ramrakhya et al. (2022) was released, a new higher-quality dataset called HM3D-Semantics v0.1 Yadav et al. (2022b) became available with 120 annotated 3D scenes, and a few months after that HM3D-Semantics v0.2 added 96 additional scenes. Scaling Ramrakhya *et al*.'s approach to continuously incorporate new scenes involves replicating that entire effort again and again.

On the other hand, training with reinforcement learning (RL) is trivially scalable once annotated 3D scans are available. However, as demonstrated in Maksymets *et al*. Maksymets et al. (2021), RL requires careful reward engineering, the reward function typically used for OBJECTNAV actually *penalizes* exploration (even though the task requires it), and the existing RL policies overfit to the small number of available environments.

Our primary technical contribution is PIRLNav, an approach for pretraining with BC and finetuning with RL for OBJECTNAV. BC pretrained policies provide a reasonable starting point for 'bootstrapping' RL and make the optimization easier than learning from scratch. In fact, we show that BC

pretraining even unlocks RL with sparse rewards. Sparse rewards are simple (do not involve any reward engineering) and do not suffer from the unintended consequences described above. However, learning from scratch with sparse rewards is typically out of reach since most random action trajectories result in no positive rewards.

While combining IL and RL has been studied in prior work Schaal (1996); Das et al. (2018); Rajeswaran et al. (2018); Baker et al. (2022); Gupta et al. (2019), the main technical challenge in the context of modern neural networks is that imitation pretraining results in weights for the policy (or actor), but not a value function (or critic). Thus, naively initializing a new RL policy with these BC-pretrained policy weights often leads to catastrophic failures due to destructive policy updates early on during RL training, especially for actor-critic RL methods Uchendu et al. (2022). To overcome this challenge, we present a two-stage learning scheme involving a critic-only learning phase first that gradually transitions over to training both the actor and critic. We also identify a set of practical recommendations for this recipe to be applied to OBJECTNAV. This leads to a PIRLNav policy that advances the state-the-art on OBJECTNAV from $60.0\%$ success rate (in Chaplot et al. (2020)) to $65.0\%$ ($+5.0\%$, $8.3\%$ relative improvement).

Next, using this BC→RL training recipe, we conduct an empirical analysis of design choices. Specifically, an ingredient we investigate is whether human demonstrations can be replaced with 'free' (automatically generated) sources of demonstrations for OBJECTNAV, *e.g.* (1) shortest paths (SP) between the agent's start location and the closest object instance, or (2) task-agnostic frontier exploration Yamauchi (1997) (FE) of the environment followed by shortest path to goal-object upon observing it. We ask and answer the following:

1. *'Do human demonstrations capture any unique OBJECTNAV-specific behaviors that shortest paths and frontier exploration trajectories do not?'* Yes. We find that BC / BC→RL on human demonstrations outperforms BC / BC→RL on shortest paths and frontier exploration trajectories respectively. When we control the number of demonstrations from each source such that BC success on TRAIN is the same, RL-finetuning when initialized from BC on human demonstrations still outperforms the other two.
2. *'How does performance after RL scale with BC dataset size?'* We observe diminishing returns from RL-finetuning as we scale BC dataset size. This suggests, by effectively leveraging the trade-off curve between size of pretraining dataset size *vs.* performance after RL-Finetuning, we can achieve closer to state-of-the-art results without investing into a large dataset of BC demonstrations.
3. *'Does BC on frontier exploration demonstrations present similar scaling behavior as BC on human demonstrations?'* No. We find that as we scale frontier exploration demonstrations past $70k$ trajectories, the performance plateaus.

Finally, we present an analysis of the failure modes of our OBJECTNAV policies and present a set of guidelines for further improving them. Our policy's primary failure modes are: a) Dataset issues: comprising of missing goal annotations, and navigation meshes blocking the path, b) Navigation errors: primarily failure to navigate between floors, c) Recognition failures: where the agent does not identify the goal object during an episode, or confuses the specified goal with a semantically-similar object.

## 2 RELATED WORK

**ObjectGoal Navigation**. Prior works on OBJECTNAV have used end-to-end RL Mousavian et al. (2019); Ye et al. (2021); Maksymets et al. (2021), modular learning Chaplot et al. (2020); Liang et al. (2021); Ramakrishnan et al. (2022), and imitation learning Ramrakhya et al. (2022); Yadav et al. (2022a). Works that use end-to-end RL have proposed improved visual representations Mousavian et al. (2019); Yang et al. (2019), auxiliary tasks Ye et al. (2021), and data augmentation techniques Maksymets et al. (2021) to improve generalization to unseen environments. Improved visual representations include object relation graphs Yang et al. (2019) and semantic segmentations Mousavian et al. (2019). Ye *et al*. Ye et al. (2021) use auxiliary tasks like predicting environment dynamics, action distributions, and map coverage in addition to OBJECTNAV and achieve promising results. Maksymets *et al*. Maksymets et al. (2021) improve generalization of RL agents by training with artificially inserted objects and proposing a reward to incentivize exploration.

Modular learning methods for OBJECTNAV have also emerged as a strong competitor Chaplot et al. (2020); Liang et al. (2021); Ramakrishnan et al. (2020). These methods rely on separate modules for semantic mapping that build explicit structured map representations, a high-level semantic

exploration module that is learned through RL to solve the 'where to look?' subproblem, and a low-level navigation policy that solves 'how to navigate to $(x, y)$?'.

The current state-of-the-art methods on OBJECTNAV Ramrakhya et al. (2022); Yadav et al. (2022a) make use of BC on a large dataset of $80k$ human demonstrations. with a simple CNN+RNN policy architecture. In this work, we improve on them by developing an effective approach to finetune these imitation-pretrained policies with RL.

**Imitation Learning and RL Finetuning**. Prior works have considered a special case of learning from demonstration data. These approaches initialize policies trained using behavior cloning, and then fine-tune using on-policy reinforcement learning Schaal (1996); Rajeswaran et al. (2018); Baker et al. (2022); Gupta et al. (2019); Peters & Schaal (2008); Kober & Peters (2008), On classical tasks like cart-pole swing-up Schaal (1996), balance, hitting a baseball Peters & Schaal (2008), and underactuated swing-up Kober & Peters (2008), demonstrations have been used to speed up learning by initializing policies pretrained on demonstrations for RL. Similar to these methods, we also use a on-policy RL algorithm for finetuning the policy trained with behavior cloning. Rajeswaran *et al*. Rajeswaran et al. (2018) (DAPG) pretrain a policy using behavior cloning and use an augmented RL finetuning objective to stay close to the demonstrations which helps reduce sample complexity. Unfortunately DAPG is not feasible in our setting as it requires solving a systems research problem to efficiently incorporate replaying demonstrations and collecting experience online at our scale. Rajeswaran et al. (2018) show results of the approach on a dexterous hand manipulation task with a small number of demonstrations that can be loaded in system memory and therefore did not need to solve this system challenge. This is not possible in our setting, just the $256 \times 256$ RGB observations for the $77k$ demos we collect would occupy over 2 TB memory, which is out of reach for all but the most exotic of today's systems. There are many methods for incorporating demonstrations/imitation learning with off-policy RL Nair et al. (2020); Lu et al. (2021); Kalashnikov et al. (2018); Peng et al. (2019); Wang et al. (2018). Unfortunately these methods were not designed to work with recurrent policies and adapting off-policy methods to work with recurrent policies is challenging Kapturowski et al. (2019). See the Appendix A for more details. The RL finetuning approach that demonstrates results with an actor-critic and high-dimensional visual observations, and is thus most closely related to our setup is proposed in VPT Baker et al. (2022). Their approach uses Phasic Policy Gradients (PPG) Cobbe et al. (2020) with a KL-divergence loss between the current policy and the frozen pretrained policy, and decays the KL loss weight $\rho$ over time to enable exploration during RL finetuning. Our approach uses Proximal Policy Gradients (PPO) Schulman et al. (2017) instead of PPG, and therefore does not require a KL constraint, which is compute-expensive, and performs better on OBJECTNAV.

## 3 OBJECTNAV AND IMITATION LEARNING

### 3.1 OBJECTNAV

In OBJECTNAV an agent is tasked with searching for an instance of the specified object category (*e.g.*, 'bed') in an unseen environment. The agent must perform this task using only egocentric perceptions. Specifically, a RGB camera, Depth sensor[2], and a GPS+Compass sensor that provides location and orientation relative to the start position of the episode. The action space is discrete and consists of MOVE_FORWARD $(0.25m)$, TURN_LEFT $(30°)$, TURN_RIGHT $(30°)$, LOOK_UP $(30°)$, LOOK_DOWN $(30°)$, and STOP actions. An episode is considered successful if the agent stops within $1m$ Euclidean distance of the goal object within 500 steps and is able to view the object by taking turn actions Batra et al. (2020).

We use scenes from the HM3D-Semantics v0.1 dataset Yadav et al. (2022b). The dataset consists of 120 scenes and 6 unique goal object categories. We evaluate our agent using the train/val/test splits from the 2022 Habitat Challenge[3].

### 3.2 OBJECTNAV DEMONSTRATIONS

Ramrakhya *et al*. Ramrakhya et al. (2022) collected OBJECTNAV demonstrations for the Matterport3D dataset Chang et al. (2017). We begin our study by replicating this effort and collect demonstrations

---

[2]We don't use this sensor as we don't find it helpful.
[3]https://aihabitat.org/challenge/2022/

for the HM3D-Semantics v0.1 dataset Yadav et al. (2022b). We use Ramrakhya *et al.*'s Habitat-WebGL infrastructure to collect $77k$ demonstrations, amounting to $\sim 2378$ human annotation hours.

### 3.3 IMITATION LEARNING FROM DEMONSTRATIONS

We use behavior cloning to pretrain our OBJECTNAV policy on the human demonstrations we collect. Let $\pi_\theta^{BC}(a_t \mid o_t)$ denote a policy parametrized by $\theta$ that maps observations $o_t$ to a distribution over actions $a_t$. Let $\tau$ denote a trajectory consisting of state, observation, action tuples: $\tau = \left(s_0, o_0, a_0, \ldots, s_T, o_T, a_T\right)$ and $\mathcal{T} = \left\{\tau^{(i)}\right\}_{i=1}^N$ denote a dataset of human demonstrations. The optimal parameters are

$$\theta^* = \arg\min_\theta \sum_{i=1}^N \sum_{(o_t, a_t) \in \tau^{(i)}} -\log\left(\pi_\theta^{BC}(a_t \mid o_t)\right) \tag{1}$$

We use inflection weighting Wijmans et al. (2019) to adjust the loss function to upweight timesteps where actions change (*i.e.* $a_{t-1} \neq a_t$).

Our **ObjectNav policy** architecture is a simple CNN+RNN model from Yadav et al. (2022a). To encode RGB input ($i_t = \text{CNN}(I_t)$), we use a ResNet50 He et al. (2016). Following Yadav et al. (2022a), the CNN is first pre-trained on the Omnidata starter dataset Eftekhar et al. (2021) using the self-supervised pretraining method DINO Caron et al. (2021) and then finetuned during OBJECTNAV training. The GPS+Compass inputs, $P_t = (\Delta x, \Delta y, \Delta z)$, and $R_t = (\Delta\theta)$, are passed through fully-connected layers $p_t = \text{FC}(P_t), r_t = \text{FC}(R_t)$ to embed them to 32-d vectors. Finally, we convert the object goal category to one-hot and pass it through a fully-connected layer $g_t = \text{FC}(G_t)$, resulting in a 32-d vector. All of these input features are concatenated to form an observation embedding, and fed into a 2-layer, 2048-d GRU at every timestep to predict a distribution over actions $a_t$ - formally, given current observations $o_t = [i_t, p_t, r_t, g_t]$, $(h_t, a_t) = \text{GRU}(o_t, h_{t-1})$. To reduce overfitting, we apply color-jitter and random shifts Yarats et al. (2021) to the RGB inputs.

## 4 RL FINETUNING

Our motivation for RL-finetuning is two-fold. First, finetuning may allow for higher performance as behavior cloning is known to suffer from a train/test mismatch – when training, the policy sees the result of taking ground-truth actions, while at test-time, it must contend with the consequences of its own actions. Second, collecting more human demonstrations on new scenes or simply to improve performance is time-consuming and expensive. On the other hand, RL-finetuning is trivially scalable (once annotated 3D scans are available) and has the potential to reduce the amount of human demonstrations needed.

### 4.1 SETUP

The RL objective is to find a policy $\pi_\theta(a|s)$ that maximizes expected sum of discounted future rewards. Let $\tau$ be a sequence of object, action, reward tuples $(o_t, a_t, r_t)$ where $a_t \sim \pi_\theta(\cdot \mid o_t)$ is the action sampled from the agent's policy, and $r_t$ is the reward. For a discount factor $\gamma$, the optimal policy is

$$\pi^* = \arg\max_\pi \mathbb{E}_{\tau \sim \pi}[R_T], \text{ where } R_T = \sum_{t=1}^T \gamma^{t-1} r_t. \tag{2}$$

To solve this maximization problem, actor-critic RL methods learn a state-value function $V(s)$ (also called a critic) in addition to the policy (also called an actor). The critic $V(s_t)$ represents the expected value of returns $R_t$ when starting from state $s_t$ and acting under the policy $\pi$, where returns are defined as $R_t = \sum_{i=t}^T \gamma^{i-t} r_i$. We use DD-PPO Wijmans et al. (2020), a distributed implementation of PPO Schulman et al. (2017), an on-policy RL algorithm. Given a $\theta$-parameterized policy $\pi_\theta$ and a set of rollouts, PPO updates the policy as follows. Let $\hat{A}_t = R_t - V(s_t)$, be the advantage estimate and $p_t(\theta) = \frac{\pi_\theta(a_t|o_t)}{\pi_{\theta_{\text{old}}}(a_t|o_t)}$ be the ratio of the probability of action $a_t$ under current policy and under the policy used to collect rollouts. The parameters are updated by maximizing:

$$J^{PPO}(\theta) = \mathbb{E}_t\left[\min\left(p_t(\theta)\hat{A}_t, \text{clip}(p_t(\theta), 1 - \epsilon, 1 + \epsilon)\hat{A}_t\right)\right] \tag{3}$$

We use a sparse success reward. Sparse success is simple (does not require hyperparameter optimization) and has fewer unintended consequences (*e.g.* Maksymets *et al.* Maksymets et al. (2021) showed that typical dense rewards used in OBJECTNAV actually *penalize* exploration, even though exploration is necessary for OBJECTNAV in new environments). Sparse rewards are desirable but typically difficult to use with RL (when initializing training from scratch) because they result in nearly all trajectories achieving $0$ reward, making it difficult to learn. However, since we pretrain with BC, we do not observe any such pathologies.

## 4.2 FINETUNING METHODOLOGY

We use the behavior cloned policy $\pi_\theta^{BC}$ weights to initialize the actor parameters. However, notice that during behavior cloning we do not learn a critic nor is it easy to do so – a critic learned on human demonstrations (during behavior cloning) would be overly optimistic since all it sees are successes. Thus, we must learn the critic from scratch during RL. Naively finetuning the actor with a randomly-initialized critic leads to a rapid drop in performance[4] (see figure 5) since the critic provides poor value estimates which influence the actor's gradient updates (see Eq.equation 3). We address this issue by using a two-phase training regime:

**Phase 1: Critic Learning**. In the first phase, we rollout trajectories using the frozen policy, pretrained using BC, and use them to learn a critic. To ensure consistency of rollouts collected for critic learning with RL training, we sample actions (as opposed to using `argmax` actions) from the pre-trained BC policy: $a_t \sim \pi_\theta(s_t)$. We train the critic until its loss plateaus. In our experiments, we found $8M$ steps to be sufficient. In addition, we also initialize the weights of the critic's final linear layer close to zero to stabilize training.

**Phase 2: Interactive Learning**. In the second phase, we unfreeze the actor RNN[5] and finetune both actor and critic weights. We find that naively switching from phase 1 to phase 2 leads to small improvements in policy performance at convergence. We gradually decay the critic learning rate from $2.5 \times 10^{-4}$ to $1.5 \times 10^{-5}$ while warming-up the policy learning rate from 0 to $1.5 \times 10^{-5}$ between $8M$ to $12M$ steps, and then keeping both at $1.5 \times 10^{-5}$ through the course of training. We find that using this learning rate schedule helps improve policy performance. For parameters that are shared between the actor and critic (*i.e.* the RNN), we use the lower of the two learning rates (*i.e.* always the actor's in our schedule). To summarize our finetuning methodology:

– First, we initialize the weights of the policy network with the IL-pretrained policy and initialize critic weights close to zero. We freeze the actor and shared weights. The only learnable parameters are in the critic.
– Next, we learn the critic weights on rollouts collected from the pretrained, frozen policy.
– After training the critic, we warmup the policy learning rate and decay the critic learning rate.
– Once both critic and policy learning rate reach a fixed learning rate, we train the policy to convergence.

## 4.3 RESULTS

**Comparing with the RL-finetuning approach in VPT Baker et al. (2022)**. We start by comparing our proposed RL-finetuning approach with the approach used in VPT Baker et al. (2022). Specifically, Baker et al. (2022) proposed initializing the critic weights to zero, replacing entropy term with a KL-divergence loss between the frozen IL policy and the RL policy, and decay the KL divergence loss coefficient, $\rho$, by a fixed factor after every iteration. Notice that this prevents the actor from drifting too far too quickly from the IL policy, but does not solve uninitialized critic problem. To ensure fair comparison, we implement this method within our DD-PPO framework to ensure that any performance difference is due to the fine-tuning algorithm and not tangential implementation differences. Complete training details are in the Appendix C.3. We keep hyperparameters constant for our approach for all experiments. Table 1a reports results on HM3D VAL for the two approaches using $20k$ human demonstrations. We find that PIRLNav achieves $+2.2\%$ Success compared to VPT and comparable SPL.

---

[4]After the initial drop, the performance increases but the improvements on success are small.
[5]The CNN and non-visual observation embedding layers remain frozen. We find this to be more stable.

| Method | Success (↑) | SPL (↑) |
|---|---|---|
| 1) BC | 52.0 | 20.6 |
| 2) BC→RL-FT w/ VPT | 59.7 ±0.70 | **28.6** ±0.89 |
| 3) PIRLNav (Ours) | **61.9** ±0.47 | 27.9 ±0.56 |

**(a)** Comparison with VPT on HM3D VAL Ramakrishnan et al. (2020); Yadav et al. (2022b)

| Method | Success (↑) | SPL (↑) |
|---|---|---|
| 1) BC | 52.0 | 20.6 |
| 2) BC→RL-FT | 53.6 ±1.01 | **28.6** ±0.50 |
| 3) BC→RL-FT (+ Critic Learning) | 56.7 ±0.93 | 27.7 ±0.82 |
| 4) BC→RL-FT (+ Critic Learning, Critic Decay) | 59.4 ±0.42 | 26.9 ±0.38 |
| 5) BC→RL-FT (+ Critic Learning, Actor Warmup) | 58.2 ±0.55 | 26.7 ±0.69 |
| 6) PIRLNav | **61.9** ±0.47 | 27.9 ±0.56 |

**(b)** RL-finetuning ablations on HM3D VAL Ramakrishnan et al. (2020); Yadav et al. (2022b)

**Ablations**. Next, we conduct ablation experiments to quantify the importance of each phase in our RL-finetuning approach. Table 1b reports results on the HM3D VAL split for a policy BC-pretrained on $20k$ human demonstrations and RL-finetuned for $300M$ steps, complete training details are in Appendix C.4. First, without a gradual learning transition (row 2), *i.e.* without a critic learning and LR decay phase, the policy improves by $1.6\%$ on success and $8.0\%$ on SPL. Next, with only a critic learning phase (row 3), the policy improves by $4.7\%$ on success and $7.1\%$ on SPL. Using an LR decay schedule only for the critic after the critic learning phase improves success by $7.4\%$ and SPL by $6.3\%$, and using an LR warmup schedule for the actor (but no critic LR decay) after the critic learning phase improves success by $6.2\%$ and SPL by $6.1\%$. Finally, combining everything (critic-only learning, critic LR decay, actor LR warmup), our policy improves by $9.9\%$ on success and $7.3\%$ on SPL.

| | TEST-STD | | TEST-CHALLENGE | |
|---|---|---|---|---|
| Method | Success (↑) | SPL (↑) | Success (↑) | SPL (↑) |
| 1) Stretch Chaplot et al. (2020) | 60.0% | 34.0% | 56.0% | 29.0% |
| 2) ProcTHOR-Large Deitke et al. (2022) | 54.0% | 32.0% | - | - |
| 3) Habitat-Web Ramrakhya et al. (2022) | 55.0% | 22.0% | - | - |
| 4) DD-PPO Team (2020) | 26.0% | 12.0% | - | - |
| 5) Populus A. | 66.0% | 32.0% | 60.0% | 30.0% |
| 6) ByteBOT | 68.0% | 37.0% | 64.0% | 35.0% |
| 7) PIRLNav[6] | **65.0**% | 33.0% | **65.0**% | 33.0% |

**Table 2.** Results on HM3D TEST-STANDARD and TEST-CHALLENGE Team (2020); Yadav et al. (2022b). Unpublished works submitted only to the OBJECTNAV leaderboard have been grayed out.

**ObjectNav Challenge 2022 Results**. Using our overall two-stage training approach of BC-pretraining followed by RL-finetuning, we achieve state-of-the-art results on OBJECTNAV– $65.0\%$ success and $33.0\%$ SPL on both the TEST-STANDARD and TEST-CHALLENGE splits and $70.4\%$ success and $34.1\%$ SPL on VAL. Table 2 compares our results with the top-4 entries to the Habitat OBJECTNAV Challenge 2022 Team (2020). Our approach outperforms Stretch Chaplot et al. (2020) on success rate on both TEST-STANDARD and TEST-CHALLENGE and is comparable on SPL ($1\%$ worse on TEST-STANDARD, $4\%$ better on TEST-CHALLENGE). ProcTHOR Deitke et al. (2022), which uses $10k$ procedurally-generated environments for training, achieves $54\%$ success and $32\%$ SPL on TEST-STANDARD split, which is $11\%$ worse at success and $1\%$ worse at SPL than ours. For sake of completeness, we also report results of two unpublished entries uploaded to the leaderboard – Populus A. and ByteBOT. Unfortunately, there is no associated report yet with these entries, so we are unable to comment on the details of these approaches, or even whether the comparison is meaningful.

## 5 ROLE OF DEMONSTRATIONS IN BC→RL TRANSFER

Our decision to use human demonstrations for BC-pretraining before RL-finetuning was motivated by results in prior work Ramrakhya et al. (2022). Next, we examine if other cheaper sources of demonstrations lead to equally good BC→RL generalization. Specifically, we consider 3 sources of demonstrations:

**Shortest paths (SP)**. These demonstrations are generated by greedily sampling actions to fit the geodesic shortest path to the nearest navigable goal object, computed using the ground-truth map of

---

[6]The approach is called "BadSeed" on the HM3D leaderboard: eval.ai/web/challenges/challenge-page/1615/leaderboard/3899

| Training demonstrations | Success (↑) | SPL (↑) |
|---|---|---|
| Shortest paths ($240k$) | $6.4\%$ | $5.0\%$ |
| Frontier exploration ($70k$) | $44.9\%$ | $21.5\%$ |
| Human demonstrations ($77k$) | $\mathbf{64.1\%}$ | $\mathbf{27.1\%}$ |

**(a)** Performance on HM3D VAL with imitation learning on SP, FE, and HD demonstrations. The size of each demonstration dataset is picked such that total steps of experience is $\sim 12M$.

| Training demonstrations | BC Success (↑) | RL-FT Success (↑) |
|---|---|---|
| 1) SP | $\mathbf{5.2\%}$ | $34.8\%$ |
| 2) HD | $0.0\%$ | $\mathbf{57.2\%}$ |
| 3) FE | $\mathbf{26.3\%}$ | $43.0\%$ |
| 4) HD | $0.0\%$ | $\mathbf{57.2\%}$ |

**(b)** Results on SP-favoring and FE-Favoring splits.

the environment. These demonstrations do not capture any exploration, they only capture success at the OBJECTNAV task via the most efficient path.

**Task-Agnostic Frontier Exploration (FE)** Chaplot et al. (2020). These are generated by using a 2-stage approach: 1) Exploration: where a task-agnostic strategy is used to maximize exploration coverage and build a top-down semantic map of the environment, and 2) Goal navigation: once the goal object is detected by the semantic predictor, the developed map is used to reach it by following the shortest path. These demonstrations capture OBJECTNAV-agnostic exploration.

**Human Demonstrations (HD)** Ramrakhya et al. (2022). These are collected by asking humans on Mechanical Turk to control an agent and navigate to the goal object. Humans are provided access to the first-person RGB view of the agent and tasked to reach within 1m of the goal object category. These demonstrations capture human-like OBJECTNAV-specific exploration.

## 5.1 RESULTS WITH BEHAVIOR CLONING

Using the BC setup described in Sec. 3.3, we train on SP, FE, and HD demonstrations. Since these demonstrations vary in trajectory length (*e.g.* SP are significantly shorter than FE), we collect $\sim 12M$ steps of experience with each method. That amounts to $240k$ SP, $70k$ FE, and $77k$ HD demonstrations respectively. As shown in Table 3a, BC on $240k$ SP demonstrations leads to $6.4\%$ success and $5.0\%$ SPL. We believe this poor performance is due to an imitation gap Weihs et al. (2021a), *i.e.* the shortest path demonstrations are generated with access to privileged information (ground-truth map of the environment) which is not available to the policy during training. Without a map, following the shortest path in a new environment to find a goal object is not possible. BC on $70k$ FE demonstrations achieves $44.9\%$ success and $21.5\%$ SPL, which is significantly better than BC on shortest paths ($+38.5\%$ success, $+16.5\%$ SPL). Finally, BC on $77k$ HD obtains the best results – $64.1\%$ success, $27.1\%$ SPL. These trends suggest that task-specific exploration (captured in human demonstrations) leads to much better generalization than task-agnostic exploration (FE) or shortest paths (SP).

## 5.2 RESULTS WITH RL FINETUNING

Using the BC-pretrained policies on SP, FE, and HD demonstrations as initialization, we RL-finetune each using our approach described in Sec. 4. These results are summarized in figure 3a. Perhaps intuitively, the trends after RL-finetuning follow the same ordering as BC-pretraining, *i.e.* RL-finetuning from BC on HD > FE > SP. But there are two factors that could be leading to this ordering after RL-finetuning – 1) inconsistency in performance at initialization (*i.e.* BC on HD is already better than BC on FE), and 2) amenability of each of these initializations to RL-finetuning (*i.e.* is RL-finetuning from HD init better than FE init?).

We are interested in answering (2), and so we control for (1) by selecting BC-pretrained policy weights across SP, FE, and HD that have equal performance on a subset of TRAIN = $\sim 48.0\%$ success. This essentially amounts to selecting BC-pretraining checkpoints for FE and HD from earlier in training as $\sim 48.0\%$ success is the maximum for SP.

figure 3b shows the results after BC and RL-finetuning on a subset of the HM3D TRAIN and on HM3D VAL. First, note that at BC-pretraining TRAIN success rates are equal (= $\sim 48.0\%$), while on VAL FE is slightly better than HD followed by SP. We find that after RL-finetuning, the policy trained on HD still leads to higher VAL success ($66.1\%$) compared to FE ($51.3\%$) and SP ($43.6\%$). Notice that RL-finetuning from SP leads to high TRAIN success, but low VAL success, indicating significant overfitting. FE has smaller TRAIN-VAL gap after RL-finetuning but both are worse than HD, indicating underfitting. These results show that learning to imitate human demonstrations equips the agent with navigation strategies that enable better RL-finetuning generalization compared to imitating other kinds of demonstrations, even when controlled for the same BC-pretraining accuracy.

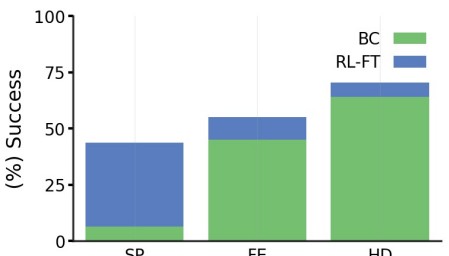
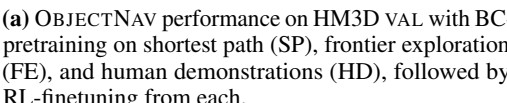
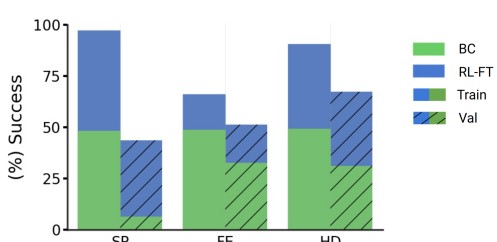

**(a)** OBJECTNAV performance on HM3D VAL with BC-pretraining on shortest path (SP), frontier exploration (FE), and human demonstrations (HD), followed by RL-finetuning from each.

**(b)** BC and RL performance for shortest paths (SP), frontier exploration (FE), and human demonstrations (HD) with equal BC training success on HM3D TRAIN (left) and VAL (right).

**Results on SP-favoring and FE-favoring episodes**. To further emphasize that imitating human demonstrations is key to good generalization, we created two subsplits from the HM3D VAL split that are adversarial to HD performance – SP-favoring and FE-favoring. The SP-favoring VAL split consists of episodes where BC on SP achieved a higher performance compared to BC on HD, *i.e.* we select episodes where BC on SP succeeded but BC on HD did not or both BC on SP and BC on HD failed. Similarly, we also create an FE-favoring VAL split using the same sampling strategy biased towards BC on FE. Next, we report the performance of RL-finetuned from BC on SP, FE, and HD on these two evaluation splits in Table 3b. On both SP-favoring and FE-favoring, BC on HD is at $0\%$ success (by design), but after RL-finetuning, is able to significantly outperform RL-finetuning from the respective BC on SP and FE policies.

## 5.3 SCALING LAWS OF BC AND RL

In this section, we investigate how BC-pretraining $\rightarrow$ RL-finetuning success scales with no. of BC demonstrations.

**Human demonstrations**. We create HD subsplits ranging in size from $2k$ to $77k$ episodes, and BC-pretrain policies with the same set of hyperparameters on each split. Then, for each, we RL-finetune from the best-performing checkpoint. The resulting BC and RL success on HM3D VAL *vs.* no. of HD episodes is plotted in figure 1. Similar to Ramrakhya et al. (2022), we see promising scaling behavior with more BC demonstrations.

Interestingly, as we increase the size of of the BC pretraining dataset and get to high BC accuracies, the improvements from RL-finetuning decrease. *E.g.* at $20k$ BC demonstrations, the BC$\rightarrow$RL improvement is $10.1\%$ success, while at $77k$ BC demonstrations, the improvement is $6.3\%$. Furthermore, with $35k$ BC-pretraining demonstrations, the RL-finetuned success is only $4\%$ worse than RL-finetuning from $77k$ BC demonstrations ($66.4\%$ *vs.* $70.4\%$). Both suggest that by effectively leveraging the trade-off between the size of the BC-pretraining dataset *vs.* performance gains after RL-finetuning, it may be possible to achieve close to state-of-the-art results without large investments in demonstrations.

**How well does FE Scale?** In Section 5.1, we showed that BC on human demonstrations outperforms BC on both shortest paths and frontier exploration demonstrations, when controlled for the same amount of training experience. In contrast to human demonstrations however, collecting shortest paths and frontier exploration demonstrations is cheaper, which makes scaling these demonstration datasets easier. Since BC performance on shortest paths is significantly worse even with 3x more demonstrations compared to FE and HD ($240k$ SP *vs.* $70k$ FE and $77k$ HD demos, Sec. 5.1), we focus on scaling FE demonstrations. figure 4 (a) plots performance on HM3D VAL against FE dataset size and a curve fitted using $75k$ demonstrations to predict performance on FE dataset-sizes $\geq 75k$. We created splits ranging in size from $10k$ to $150k$. Increasing the dataset size doesn't consistently improve performance and saturates after $70k$ demonstrations, suggesting that generating more FE demonstrations is unlikely to help. We hypothesize that the saturation is because these demonstrations don't capture task-specific exploration.

## 6 FAILURE MODES

To better understand the failure modes of our BC$\rightarrow$RL OBJECTNAV policies, we manually annotate 592 failed HM3D VAL episodes from our best OBJECTNAV agent. See figure 4 (b). The most

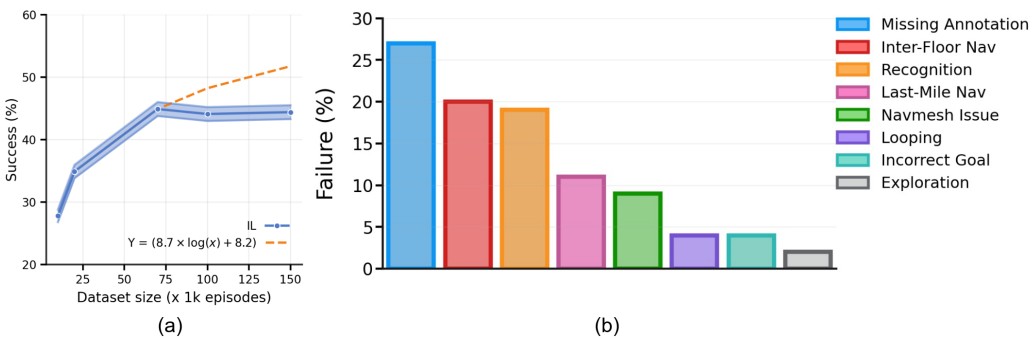

**Figure 4.** (a) Success on ObjectNav HM3D VAL split vs. no. of frontier exploration demonstrations for training. (b) Failure modes of our best BC→RL OBJECTNAV policy

common failure modes are:

**Missing Annotations** (27%): Episodes where the agent navigates to the correct goal object category but the episode is counted as a failure due to missing annotations in the data.

**Inter-Floor Navigation** (21%): The object is on a different floor and the agent fails to climb up/down the stairs.

**Recognition Failure** (20%): The agent sees the object in its field of view but fails to navigate to it.

**Last Mile Navigation** Wasserman et al. (2022) (12%). Repeated collisions against objects or mesh geometry close to the goal object preventing the agent from reaching close to it.

**Navmesh Failure** (9%). Hard-to-navigate meshes blocking the path of the agent. *E.g.* in one instance, the agent fails to climb stairs because of a narrow nav mesh on the stairs.

**Looping** (4%). Repeatedly visiting the same location and not exploring the rest of the environment.

**Semantic Confusion** (5%). Confusing the goal object with a semantically-similar object. *E.g.* 'armchair' for 'sofa'.

**Exploration Failure** (2%). Catch-all for failures in a complex navigation environment, early termination, semantic failures (*e.g.* looking for a chair in a bathroom), *etc.*

As can be seen in figure 4 (b), most failures (∼36%) are due to issues in the OBJECTNAV dataset – 27% due to missing object annotations + 9% due to holes / issues in the navmesh. 21% failures are due to the agent being unable to climb up/down stairs. We believe this happens because climbing up / down stairs to explore another floor is a difficult behavior to learn and there are few episodes that require this. Oversampling inter-floor navigation episodes during training can help with this. Another failure mode is failing to recognize the goal object – 20% where the object is in the agent's field of view but it does not navigate to it, and 5% where the agent navigates to another semantically-similar object. Advances in the visual backbone and object recognition can help address these. Prior works Ramrakhya et al. (2022); Chaplot et al. (2020) have used explicit semantic segmentation modules to recognize objects at each step of navigation. Incorporating this within the BC→RL training pipeline could help. 11% failures are due to last mile navigation, suggesting that equipping the agent with better goal-distance estimators could help. Finally, only ∼6% failures are due to looping and lack of exploration, which is promising!

## 7 CONCLUSION

To conclude, we propose PIRLNav, an approach to combine imitation using behavior cloning (BC) and reinforcement learning (RL) for OBJECTNAV, wherein we pretrain a policy with BC on $77k$ human demonstrations and then finetune it with RL, leading to state-of-the-art results on OBJECTNAV (65% success, 5% improvement over previous best). Next, using this BC→RL training recipe, we present a thorough empirical study of the impact of different demonstration datasets used for BC-pretraining on downstream RL-finetuning performance. We show that BC / BC→RL on human demonstrations outperforms BC / BC→RL on shortest paths and frontier exploration trajectories, even when we control for same BC success on TRAIN. We also show that as we scale the pretraining dataset size for BC and get to higher BC success rates, the improvements from RL-finetuning start to diminish. Finally, we characterize our agent's failure modes, and find that the largest sources of error are 1) dataset annotation noise, and inability of the agent to 2) navigate across floors, and 3) recognize the correct goal object.

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

## A  PRIOR WORK IN RL FINETUNING

### A.1  DAPG RAJESWARAN ET AL. (2018)

**Preliminaries**. Rajeswaran *et al*. Rajeswaran et al. (2018) proposed DAPG, a method which incorporates demonstrations in RL, and thus quite relevant to our methodology. DAPG first pretrains a policy using behavior cloning then finetunes the policy using an augmented RL objective (shown in Eq. equation 4). DAPG proposes to use different parts of demonstrations dataset during different stages of learning for tasks involving sequence of behaviors. To do so, they add an additional term to the policy gradient objective:

$$g_{aug} = \sum_{(s,a) \in \tau \sim \pi_\theta} \nabla_\theta \log_{\pi_\theta}(a|s) A^\pi(s,a) +$$

$$\sum_{(s,a) \in \tau \sim \mathcal{T}} \nabla_\theta \log_{\pi_\theta}(a|s) w(s,a) \quad (4)$$

Here $\tau \sim \pi_\theta$ is a trajectory obtained by executing the current policy, $\tau \sim \mathcal{T}$ denotes a trajectory obtained by replaying a demonstration, and $w(s,a)$ is a weighting function to alternate between imitation and reinforcement learning. DAPG uses a heuristic weighting scheme to set $w(s,a)$ to decay the auxiliary objective:

$$w(s,a) = \lambda_0 \lambda_1^k \max_{(s',a') \in \tau \sim \pi_\theta} A^{\pi_\theta}(s',a') \forall (s,a) \quad (5)$$

where $\lambda_0$ and $\lambda_1$ are hyperparameters and $k$ is the update iteration counter. The decaying weighting term $\lambda_1^k$ is used to avoid biasing the gradient towards the demonstrations data towards the end of training.

**Implementation Details**. Rajeswaran et al. (2018) showed results of using DAPG on dexterous hand manipulation tasks for object relocation, in-hand manipulation, tool use, *etc*. To train the policy with behavior cloning, they use 25 demonstrations for each task gathered using the Mujoco HAPTIX system Kumar & Todorov (2015). The small size of the demonstrations dataset and the observation input allows DAPG to load the demonstrations dataset in system memory which makes it feasible to compute the augmented RL objective shown above.

**Challenges in adopting Rajeswaran et al. (2018)'s setup**. Compared to Rajeswaran et al. (2018), our setup uses high-dimensional visual input ($256 \times 256$ RGB observations) and $77k$ OBJECTNAV demonstrations for training. Following DAPG's training implementation, storing the visual inputs for $77k$ demonstrations in system memory would require 2TB, which is significantly higher than what is possible on today's systems. An alternative is to leverage on-the-fly demonstration replay during RL training. However, efficiently incorporating demonstration replay with experience collection online requires solving a systems research problem. Naively switching between online experience collection using the current policy and replay demonstrations would require 2x the current experience collection time, overall hurting the training throughput.

### A.2  FEASIBILITY OF OFF-POLICY RL FINETUNING

There are several methods for incorporating demonstrations with off-policy RL Nair et al. (2020); Lu et al. (2021); Kalashnikov et al. (2018); Peng et al. (2019); Wang et al. (2018). Algorithm 1 shows the general framework of off-policy RL (finetuning) methods.

Unfortunately, most of these methods use feedforward state encoders, which is ill-posed for partially observable settings. In partially observable settings, the agent requires a state representation that combines information about the state-action trajectory so far with information about the current observation, which is typically achieved using a recurrent network.

To train a recurrent policy in an off-policy setting, the full state-action trajectories need to be stored in a replay buffer to use for training, including the hidden state $h_t$ of the RNN. The policy update

---

**Algorithm 1** General framework of off-policy RL algorithm

---

**Require:** $\pi_\theta$ : Policy, $B$: replay buffer, $N$: Rounds, $I$: Policy Update Iterations
    **for** $k = 1$ to $N$ **do**
        Trajectory $\tau \leftarrow$ Rollout $\pi_\theta(\cdot|s)$ to collect trajectory $\{(s_1, a_1, r_1, h_1), ......, (s_T, a_T, r_T, h_T)\}$
        $B \leftarrow \{B\} \cup \{\tau\}$
        $\pi_\theta \leftarrow$ TrainPolicy($\pi_\theta$, $B$) for $I$ iterations
    **end for**

---

requires a sequence input for multiple time steps $\left[(s_t, a_t, r_t, h_t), ......, (s_{t+l}, a_{t+l}, r_{t+l}, h_{t+l})\right] \sim \tau$ where $l$ is sampled sequence length. Additionally, it is not obvious how the hidden state should be initialized for RNN updates when using a sampled sequence in the off-policy setting. Prior work DRQNHausknecht & Stone (2015) compared two training strategies to train a recurrent network from replayed experience:

1. **Bootstrapped Random Updates**. The episodes are sampled randomly from the replay buffer and the policy updates begin at random steps in an episode and proceed only for the unrolled timesteps. The RNN initial state is initialized to zero at the start of the update. Using randomly sampled experience better adheres to DQN's Mnih et al. (2013) random sampling strategy, but, as a result, the RNN's hidden state must be initialized to zero at the start of each policy update. Using zero start state allows for independent decorrelated sampling of short sequences which is important for robust optimization of neural networks. Although this can help RNN to learn to recover predictions from an initial state that mismatches with the hidden state from the collected experience but it might limit the ability of the network to rely on it's recurrent state and exploit long term temporal correlations.
2. **Bootstrapped Sequential Updates**. The full episode replays are sampled randomly from the replay buffer and the policy updates begin at the start of the episode. The RNN hidden state is carried forward throughout the episode. Eventhough this approach avoids the problem of finding the correct initial state it still has computational issues due to varying sequence length for each episode, and algorithmic issues due to high variance of network updates due to highly correlated nature of the states in the trajectory.

Even though using bootstrapped random updates with zero start states performed well in Atari which is mostly fully observable, R2D2Kapturowski et al. (2019) found using this strategy prevents a RNN from learning long-term dependencies in more memory critical environments like DMLab. Kapturowski et al. (2019) proposed two strategies to train recurrent policies with randomly samples sequences:

1. **Stored State**. In this strategy, the hidden state is stored at each step in the replay and use it to initialize the network at the time of policy updates. Using stored state partially remedies the issues with initial recurrent state mismatch in zero start state strategy but it suffers from 'representational drfit' leading to 'recurrent state staleness', as the stored state generated by a sufficiently old network could differ significantly from a state from the current policy.
2. **Burn-in**. In this strategy the initial part of the replay sequence is used to unroll the network and produce a start state ('burn-in period') and update the network on the remaining part of the sequence.

While R2D2 Kapturowski et al. (2019) found a combination of these strategies to be effective at mitigating the representational drift and recurrent state staleness, this increases computation and requires careful tuning of the replay sequence length $m$ and burn-in period $l$.

Both Kapturowski et al. (2019); Hausknecht & Stone (2015) demonstrate the issues associated with using a recurrent policy in an off-policy setting and present approaches that mitigate issues to some extent. Applying these techniques for Embodied AI tasks and off-policy RL finetuning is an open research problem and requires empirical evaluation of these strategies.

## B  PRIOR WORK IN IMITATION LEARNING

In Imitation Learning (IL), we use demonstrations of successful behavior to learn a policy that imitates the expert (demonstrator) providing these trajectories. The simplest approach to IL is behavior cloning (BC), which uses supervised learning to learn a policy to imitate the demonstrator. However, BC

| Parameter | Value |
|---|---|
| Number of GPUs | 64 |
| Number of environments per GPU | 8 |
| Rollout length | 64 |
| Number of mini-batches per epoch | 2 |
| Optimizer | Adam |
|    Learning rate | $1.0 \times 10^{-3}$ |
|    Weight decay | 0.0 |
|    Epsilon | $1.0 \times 10^{-5}$ |
| DDPIL sync fraction | 0.6 |

**Table 4.** Hyperparameters used for Imitation Learning.

suffers from poor generalization to unseen states, since the training mimics the actions and not their consequences. DAgger Ross et al. (2011) mitigates this issue by iteratively aggregating the dataset using the expert and trained policy $\hat{\pi_{i-1}}$ to learn the policy $\hat{\pi}_i$. Specifically, at each step $i$, the new dataset $D_i$ is generated by:

$$\pi_i = \beta \pi_{exp} + (1 - \beta)\hat{\pi}_{i-1} \qquad (6)$$

where, $\pi_{exp}$ is a queryable expert, and $\hat{\pi}_{i-1}$ is the trained policy at iteration $i - 1$. Then, we aggregate the dataset $D \leftarrow D \cup D_i$ and train a new policy $\hat{\pi}_i$ on the dataset $D$. Using experience collected by the current policy to update the policy for next iteration enables DAgger Ross et al. (2011) to mitigate the poor generalization to unseen states caused by BC. However, using DAgger Ross et al. (2011) in our setting is not feasible as we don't have a queryable human expert for policies being trained with human demonstrations.

Alternative approaches Ho & Ermon (2016); Bahdanau et al. (2019); Abbeel & Ng (2004); Ziebart et al. (2008); Fu et al. (2018) for imitation learning are variants of inverse reinforcement learning (IRL), which learn reward function from expert demonstrations in order to train a policy. IRL methods learn a parameterized $\mathcal{R}_\phi(\tau)$ reward function, which models the behavior of the expert and assigns a scalar reward to a demonstration. Given the reward $r_t$, a policy $\pi_\theta(a_t|s_t)$ is learned to map states $s_t$ to distribution over actions $a_t$ at each time step. The goal of IRL methods is to learn a reward function such that a policy trained to maximize the discounted sum of the learned reward matches the behavior of the demonstrator. Compared to prior works Ho & Ermon (2016); Bahdanau et al. (2019); Abbeel & Ng (2004); Ziebart et al. (2008); Fu et al. (2018), our setup uses a partially-observable setting and high-dimensional visual input for training. Following training implementation from prior works, storing visual inputs of demonstrations for reward model training would require $2TB$ system memory, which is significantly higher than what is possible on today's systems. Alternatively, efficiently replaying demonstrations during RL training with reward model learning in the loop requires solving an open systems research problem. In addition, applying these methods for tasks in a partially observable setting is an open research problem and requires empirical evaluation of these approaches.

## C    TRAINING DETAILS

### C.1    BEHAVIOR CLONING

We use a distributed implementation of behavior cloning by Ramrakhya et al. (2022) for our imitation pretraining. Each worker collects 64 frames of experience from 8 environments parallely by replaying actions from the demonstrations dataset. We then perform a policy update using supervised learning on 2 mini batches. For all of our BC experiments, we train the policy for $500M$ steps on 64 GPUs using Adam optimizer with a learning rate $1.0 \times 10^{-3}$ which is linearly decayed after each policy update. Tab. 4 details the default hyperparameters used in all of our training runs.

| Parameter | Value |
|---|---|
| Number of GPUs | 16 |
| Number of environments per GPU | 8 |
| Rollout length | 64 |
| PPO epochs | 2 |
| Number of mini-batches per epoch | 2 |
| Optimizer | Adam |
|    Weight decay | 0.0 |
|    Epsilon | $1.0 \times 10^{-5}$ |
| PPO clip | 0.2 |
| Generalized advantage estimation | True |
|    $\gamma$ | 0.99 |
|    $\tau$ | 0.95 |
| Value loss coefficient | 0.5 |
| Max gradient norm | 0.2 |
| DDPPO sync fraction | 0.6 |

**Table 5.** Hyperparameters used for RL finetuning.

## C.2 REINFORCEMENT LEARNING

To train our policy using RL we use PPO with Generalized Advantage Estimation (GAE) Schulman et al. (2016). We use a discount factor $\gamma$ of 0.99 and set GAE parameter $\tau$ to 0.95. We do not use normalized advantages. To parallelize training, we use DD-PPO with 16 workers on 16 GPUs. Each worker collects 64 frames of experience from 8 environments parallely and then performs 2 epochs of PPO update with 2 mini batches in each epoch. For all of our experiments, we RL finetune the policy for $300M$ steps. Tab. 5 details the default hyperparameters used in all of our training runs.

## C.3 RL FINETUNING USING VPT

To compare with RL finetuning approach proposed in VPT Baker et al. (2022) we implement the method in DD-PPO framework. Specifically, we initialize the critic weights to zero, replace the entropy term in PPO Schulman et al. (2017) with a KL-divergence loss between the frozen IL policy and RL policy, and decay the KL divergence loss coefficient, $\rho$, by a fixed factor after every iteration. This loss term is defined as:

$$L_{kl\_penalty} = \rho \text{KL}(\pi_\theta^{BC}, \pi_\theta) \tag{7}$$

where $\pi_\theta^{BC}$ is the frozen behavior cloned policy, $\pi_\theta$ is the current policy, and $\rho$ is the loss weighting term. Following, VPT Baker et al. (2022) we set $\rho$ to 0.2 at the start of training and decay it by 0.995 after each policy update. We use learning rate of $1.5 \times 10^{-5}$ without a learning rate decay for our VPT Baker et al. (2022) finetuning experiments.

## C.4 RL FINETUNING ABLATIONS

| Method | Success ($\uparrow$) | SPL ($\uparrow$) |
|---|---|---|
| 1) BC | 52.0 | 20.6 |
| 2) BC→RL-FT | 53.6 ±1.01 | **28.6** ±0.50 |
| 3) BC→RL-FT (+ Critic Learning) | 56.7 ±0.93 | 27.7 ±0.82 |
| 4) BC→RL-FT (+ Critic Learning, Critic Decay) | 59.4 ±0.42 | 26.9 ±0.38 |
| 5) BC→RL-FT (+ Critic Learning, Actor Warmup) | 58.2 ±0.55 | 26.7 ±0.69 |
| 6) PIRLNav | **61.9** ±0.47 | 27.9 ±0.56 |

**Table 6.** RL-finetuning ablations on HM3D VAL Ramakrishnan et al. (2020); Yadav et al. (2022b)

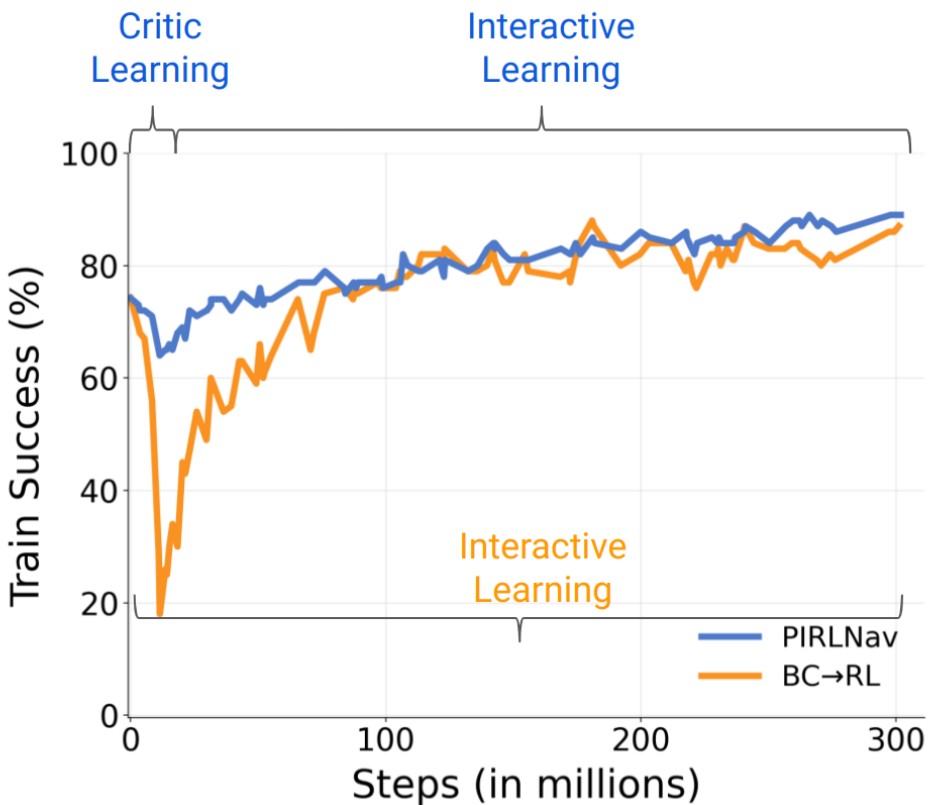

**Figure 5.** A policy pretrained on the OBJECTNAV task is used as initialization for actor weights and critic weights are initialized randomly for RL finetuning using DD-PPO. The policy performance immediately starts dropping early on during training and then recovers leading to slightly higher performance with further training.

For ablations presented in Sec. 4.3 of the main paper (also shown in Tab. 6) we use a policy pretrained on $20k$ human demonstrations using BC and finetuned for $300M$ steps using hyperparameters from Tab. 5. We try 3 learning rates ($1.5 \times 10^{-4}$, $2.5 \times 10^{-4}$, and $1.5 \times 10^{-5}$) for both BC $\rightarrow$ RL (row 2) and BC $\rightarrow$ RL (+ Critic Learning) (row 3) and we report the results with the one that works the best. For PIRLNav we use a starting learning rate of $2.5 \times 10^{-4}$ and decay it to $1.5 \times 10^{-5}$, consistent with learning rate schedule of our best performing agent. For ablations we do not tune learning rate parameters of PIRLNav, we hypothesize tuning the parameters would help improve performance.

We find BC $\rightarrow$ RL (row 2) works best with a smaller learning rate but the training performance drops significantly early on, due to the critic providing poor value estimates, and recovers later as the critic improves. See figure 5. In contrast when using proposed two phase learning setup with the learning rate schedule we do not observe a significant drop in training performance.

