# OpenReview forum: "PIRLNav: Pretraining with Imitation and RL Finetuning for ObjectNav"
_ICLR.cc/2023/Workshop/RRL — RRL 2023 Poster_

### Official Review · Reviewer_B1Bw · 2023-02-27
**An interesting study of the impact of pre-training on learning in ObjectNav problems**

**Rating:** 4
**Confidence:** 4

**Review:**

Overall evaluation:
* The paper fits very well within the framework of this workshop as it investigates how pre-training might help in ObjectNav tasks.
* Paper is written quite clearly and the appendix contains a lot of important details which help the reader understand the tested methods. The authors also clearly explain all of the preliminaries and nicely introduce the ObjectNav problem and the experiment setting.
* I would say the quality and the significance of the paper are quite high, I appreciate the numerous experiments the authors ran, as well as including some negative results (e.g. on whether frontier exploration demonstration help).

As such, I have no doubt that the paper should be presented at this workshop. Below, I attached a few comments and suggestions, but overall I think this is a good fit for this venue.

Feedback and comments:
* I find Figure 1 slightly confusing, especially the two different Y-axis scales. I think it might make more sense to show the absolute IL performance rather than plotting the difference `IL-RL`?
* Dividing fine-tuning into two phases (the first one being critic-only) seems quite interesting, but makes me think about whether we can do better. One could try to train the value function on the offline dataset with Monte Carlo rollouts. As you rightly point out, "a critic learned on human demonstrations (during behaviour cloning) would be overly optimistic since all it sees are successes". But it should be fairly easy to generate some imperfect rollouts. To be clear, I'm not asking the authors to add such an experiment, I'm only wondering whether this would strengthen the pre-training phase somehow.
* The experiment in Figure 3 is quite interesting, but I wonder what the results would look like if you start from models with equal *validation* successes. I think validation success should better reflect the overall quality of the model.
* I appreciate the comparison to VPT in the Appendix. Out of curiosity, have you tried using only the KL distillation trick without using the "set parameters of the critic to zero" trick? I'd be interested to know what is their "disentangled" impact, as KL distillation makes more sense to me than the critic zero init.
* I'm not sure if I understand Figure 8 in the Appendix. Why does the blue line performance drop (even slightly) in the first phase if we only train the critic? Shouldn't it stay exactly the same if the actor does not change?

---

### Official Review · Reviewer_Cpc2 · 2023-02-28
**A rather short (4 page) technical paper that simply omits too much detail in the main text to convincingly present their contribution and its relevance to RRL.**

**Rating:** 1
**Confidence:** 3

**Review:**

## Summary

In this work the authors present a two stage learning scheme, entitled “PIRLNav”, on the *ObjectGoal Navigation* benchmark. This algorithm has two stages: first learning with pre-training on human demonstrations, and second with RL-finetuning. They also test various other pre-training methods along with RL-finetuning, but find that their proposed configuration, with human demonstrations, is superior.

## Review

This paper presents some interesting ideas around imitation learning and RL-finetuning. However, I feel that the paper is poorly structured, lacks depth, and ultimately, is difficult to follow. It appears this *may* be because the authors have misunderstood the workshop brief —  they seemingly restricted themselves to just 4 pages for a technical paper, which was allowed to have 8 pages. After surveying their appendix, where the bulk of their work resides, I was more impressed, and their arguments were somewhat more coherent. Based purely on the main text, however, I would have to recommend *reject.* As mentioned, the quality of the paper (in 4 pages) is quite poor and, more importantly, the authors lack any clear discussion on how their work is directly relevant to RRL.

## Overall Score: 1 ⭐

Reject.

## Marking Rubric

Below I provide a breakdown of how I rate different aspects of the paper. Each category I rate out of 5.

### Relevance - 2/5 ⭐⭐

There was no direct discussion about RRL, however, the imitation learning to RL transfer can be seen as relevant to RRL. It is unfortunate that the reader must infer this link themselves. I also found the discussion on how the authors transitioned from imitation learning to RL fine tunings (including figure 6) potentially interesting for the RRL workshop, therefore it was disappointing that the section was left to the appendix.

### Novelty - 2/5 ⭐⭐

I am not convinced the work is very novel, as it primarily stitches together two extant methods.

### Significance / Importance - 2/5 ⭐⭐

I think that some of the results could be valuable to the RRL community, for example that human demonstrations were more useful than synthetic offline datasets. However, the authors lacked any kind of discussion that motivated the for the significance of their work to RRL.

### Soundness - 1/5 ⭐

The soundness of the result in the paper was one of my main concerns. Most noticeably, the results reported in the tables and figures do not include any kind of uncertainty estimates.

### Scholarship - 2/5 ⭐⭐

The authors did not cover much of the related work in the main text. Most of the related works were moved to the appendix. This is an regrettable omission.

### Clarity - 1/5 ⭐

The main text is very short and poorly structured. To name just a handful of concerns:

- In Table 1, the acronym “SPL” is never explicitly explained. What does it mean?
- Figure 1 is given on Page 1, but is only referenced on a later page — what should the reader infer from this figure?
- Moreover, I find Figure 1 itself highly confusing: what are the axis labels?; why are there two scales for the x-axis, and how are they connected? etc.
- “PIRLNav”, as the core contribution of the paper, is only mentioned in the Abstract, Intro & Conclusion — it would be helpful to have clear, dedicated discussion on it.

The clarity of this paper is improved if you read the appendix, where they seem to have put the bulk of the text. But a reader should not be *expected* to access the appendix for cogency, I feel.

### Reproducibility - 2/5 ⭐⭐

The authors provide information on the hyper-parameters used, which is appreciated. However, they do not mention how many independent runs (if any) they used for each experiment. Another way to improve the *reproducibility* score would be to share the code used in the experiments.